# Multiplexed Digital Spatial Protein Profiling Reveals Distinct Phenotypes of Mononuclear Phagocytes in Livers with Advanced Fibrosis

**DOI:** 10.3390/cells11213387

**Published:** 2022-10-26

**Authors:** Jaejun Lee, Chang Min Kim, Jung Hoon Cha, Jin Young Park, Yun Suk Yu, Hee Jung Wang, Pil Soo Sung, Eun Sun Jung, Si Hyun Bae

**Affiliations:** 1Department of Internal Medicine, Armed Forces Goyang Hospital, Goyang 10267, Korea; 2The Catholic University Liver Research Center, Department of Biomedical Science, The Graduates School of Medicine, The Catholic University of Korea, Seoul 06591, Korea; 3CbsBioscience, Inc., Daejeon 34036, Korea; 4The Catholic University Liver Research Center, Department of Biomedicine & Health Sciences, College of Medicine, The Catholic University of Korea, Seoul 06591, Korea; 5Department of Surgery, Inje University Haeundae Paik Hospital, Busan 48108, Korea; 6Division of Gastroenterology and Hepatology, Department of Internal Medicine, Seoul St. Mary’s Hospital, College of Medicine, The Catholic University of Korea, Seoul 06591, Korea; 7Department of Hospital pathology, Seoul St. Mary’s Hospital, College of Medicine, The Catholic University of Korea, Seoul 06591, Korea; 8Division of Gastroenterology and Hepatology, Department of Internal Medicine, College of Medicine, Eunpyeong St. Mary’s Hospital, The Catholic University of Korea, Seoul 03383, Korea

**Keywords:** multiplexed digital spatial protein profiling, protein signature, liver fibrosis, single-cell analysis, scar-associated macrophage

## Abstract

**Background and Aims:** Intrahepatic mononuclear phagocytes (MPs) are critical for the initiation and progression of liver fibrosis. In this study, using multiplexed digital spatial protein profiling, we aimed to derive a unique protein signature predicting advanced liver fibrosis. **Methods:** Snap-frozen liver tissues from various chronic liver diseases were subjected to spatially defined protein-based multiplexed profiling (Nanostring GeoMX^TM^). A single-cell RNA sequencing analysis was performed using Gene Expression Omnibus (GEO) datasets from normal and cirrhotic livers. **Results:** Sixty-four portal regions of interest (ROIs) were selected for the spatial profiling. Using the results from the CD68^+^ area, a highly sensitive and specific immune-related protein signature (CD68, HLA-DR, OX40L, phospho-c-RAF, STING, and TIM3) was developed to predict advanced (F3 and F4) fibrosis. A combined analysis of single-cell RNA sequencing data from GEO datasets (GSE136103) and spatially-defined, protein-based multiplexed profiling revealed that most proteins upregulated in F0–F2 livers in portal CD68^+^ cells were specifically marked in tissue monocytes, whereas proteins upregulated in F3 and F4 livers were marked in scar-associated macrophages (SAMacs) and tissue monocytes. Internal validation using mRNA expression data with the same cohort tissues demonstrated that mRNA levels for TREM2, CD9, and CD68 are significantly higher in livers with advanced fibrosis. **Conclusions:** In patients with advanced liver fibrosis, portal MPs comprise of heterogeneous populations composed of SAMacs, Kupffer cells, and tissue monocytes. This is the first study that used spatially defined protein-based multiplexed profiling, and we have demonstrated the critical difference in the phenotypes of portal MPs between livers with early- or late-stage fibrosis.

## 1. Introduction

Liver fibrosis can eventually progress to cirrhosis, which is a major concern that causes approximately 2% of deaths globally [1]. Non-resolving liver injury or liver fibrosis can cause cirrhosis, a condition that can progress to hepatocellular carcinoma. Furthermore, decompensated liver cirrhosis is accompanied by serious complications resulting from portal hypertension and/or opportunistic infections and leads to a shorter life expectancy and a deterioration in the quality of life [2]. In an effort to alleviate the burden of chronic liver disease, many studies have sought to find novel biomarkers of advanced liver fibrosis [3]. A very recent study from our group also presented a promising immune-related gene signature for predicting advanced fibrosis [4]. Although several approaches to reduce fibrosis are still under investigation, no drug has yet been demonstrated to be effective in reversing fibrosis.

Liver fibrosis involves many non-parenchymal cells (NPCs) including immune cells, endothelial cells, and mesenchymal cells [5]. These cells interplay with each other in scarring tissue, also known as the fibrotic niche. Among immune cells, intrahepatic mononuclear phagocytes (MPs) are thought to play a crucial role in exacerbating hepatic inflammation and fibrosis. The mononuclear phagocytic system (MPS) is comprised of circulating monocytes, dendritic cells, and tissue-resident macrophages, also known as Kupffer cells (KC) in the liver [6]. KCs, which reside in hepatic sinusoids, are ontogenically different from other circulating monocytes. Originating from the embryonic yolk sac, they are capable of self-renewing independently of circulating monocytes [7]. KCs play an important role in innate immunity as gatekeepers and drive the inflammatory response to liver injury. By contrast, circulating monocytes are likely dispensable for replenishing intrahepatic macrophages in homeostasis. However, in the setting of hepatic inflammation, a massive infiltration of monocyte-derived macrophages (MoMFs) to the liver is triggered [8]. MoMFs, which originate from bone marrow, are recruited from the portal triad to the injured liver by cytokines or chemokines secreted by activated KCs, including IL-1β, tumor necrosis factor α, CCL2, and CCL5 [9]. As the inflammation progresses, MoMFs differentiate into pro-fibrogenic TREM^+^ CD9^+^ scar-associated macrophages (SAMacs) and contribute to the scarring process, which leads to liver fibrosis [10]. SAMacs are reported to play a critical role in the initiation, progression, and/or resolution of liver fibrosis and continue to play a role in the fibrotic liver while fibrosis proceeds [11,12]. Overall, these findings suggest that MPs play a critical role in the process of liver fibrosis, thus it is feasible to find out biomarkers based on the MPs-associated transcriptomes.

Recently, single-cell RNA sequencing (scRNA-seq) has improved our understanding of cellular diversity and function in liver diseases. Hence, an atlas of liver NPCs has been proposed in various studies by applying single-cell RNA-sequencing in vitro and in vivo [13,14,15,16]. Ramachandran et al. reported a discrepancy in NPC phenotypes between normal liver cells and cirrhotic liver cells from humans [12]. In addition to the contribution of single-cell analysis in the field of molecular biology, spatial transcriptomics technology has also enhanced perspectives in intrahepatic molecular biology. Digital spatial profiling (DSP) (Nanostring GeoMX^TM^) is a spatially-defined protein-based multiplexed profiling, which has recently been developed for detecting RNA and/or proteins in regions of interest (ROIs) [17]. DSP is capable of detecting single cell sensitivity within a ROI at a protein level using an antibody readout and has been used in various cancer-based studies [18]. Recently, a study was conducted using a combined modality for Nanostring GeoMX^TM^ DSP and gene expression analysis, which yielded better predictive values for detecting the response rate to immunotherapy for melanoma and opened up new possibilities for this field [19].

Despite these technological advances, no study has explored the phenotypes of intrahepatic portal MPs in different stages of liver fibrosis. Here, using such a combined modality, namely multiplexed digital spatial profiling and single-cell RNA-sequencing, we aimed to identify the phenotypes of portal MPs according to the fibrosis stages and find out the potential protein biomarkers for advanced fibrosis.

## 2. Materials and Methods

### 2.1. Patients

This study included non-tumor liver tissues from 83 patients who underwent surgical resection for hepatocellular carcinoma. All tissues were obtained between 1996 and 2015 at the Ajou Medical Center (AjouMC) in Suwon, Republic of Korea. The inclusion criteria were as follows: (1) non-tumor liver tissues with chronic liver diseases and (2) tissues collected during surgical procedures, such as liver resection. Chronic liver disease was defined according to the recently introduced EASL guideline [20] as follows: a decreased liver function caused by chronic inflammation from any source, including chronic hepatitis B and C, alcoholic and non-alcoholic fatty liver disease, and other etiologies that can cause chronic liver inflammation. Chronic hepatitis B (CHB) was defined as the presence of hepatitis B surface antigens for more than 6 months, and a chronic hepatitis C (CHC) infection was defined as the presence of HCV RNA for more than 6 months. The retrospective study protocol was approved by the Institutional Review Boards of Ajou Medical Center (AJIRB-BMR-KSP-18-444) and The Catholic University of Korea (XC20EEDI0034). The fibrosis stages of every tissue sample enrolled in this study were determined by one pathologist (E.S.J), using the METAVIR scoring system.

### 2.2. RNA Extraction and Gene Expression Assay

Total RNA was extracted from the liver tissues using a RNeasy Mini Kit (QIAGEN, Hilden, Germany) with DNase I treatment (QIAGEN). Gene expression profiles were analyzed using nCounter MAX (NanoString Technologies, Seattle, WA, USA). The nCounter PanCancer Immune Profiling Panel (NanoString Technologies) was used for gene set profiling, as previously described [4].

### 2.3. Tissue Microarray (TMA) Construction

TMAs were constructed using 2-mm diameter cores punched from formalin-fixed, paraffin-embedded (FFPE) blocks. The TMA blocks were sectioned 5-μm-thick. Six TMA slides were constructed with 15 cores placed in a 5 × 3 arrangement on each slide.

### 2.4. Digital Spatial Profiling (DSP) Assay

Protein expression profiles were analyzed using GeoMx DSP (NanoString Technologies, Seattle, WA, USA). The TMA slides were stained with a mixture of detection and morphological markers. Morphological markers included Syto13 for nuclei, CD68 for macrophages, CD3 for T cells, and alpha-SMA for smooth muscle. The detection antibodies comprised one core panel and six modules of the GeoMx assay (GeoMx immune cell profiling panel, GeoMx io drug target module, GeoMx immune activation status module, GeoMx immune cell typing module, GeoMx pan-tumor module, GeoMx cell death module, and GeoMx MAPK signaling module). In total, 88 ROIs were selected around the portal tract. Each ROI was divided into CD68^+^, CD3^+^, and SMA^+^ areas. Probes attached to the detection antibodies were collected sequentially from the CD68^+^, CD3^+^, and SMA^+^ areas.

### 2.5. Selecting ROIs for Establishing the Protein Signatures

CD3 expression levels were determined using the fluorescence intensity observed in the GeoMx analysis and the median value was used to classify the ROIs into “inflammatory” and “non-inflammatory”. We excluded inflammatory ROIs because the phenotypes of the immune cells may not actually reflect the fibrogenesis process but the liver injury process.

### 2.6. Analysis of DSP Data and Validation of the Protein Signatures

The left part of the Figure 1 shows a flowchart of protein signature development. Differentially expressed proteins were analyzed by comparing fibrosis stages 0, 1, 2, 3, and 4 in CD68^+^ areas of samples. Of the differentially expressed proteins, those that were significantly different in the logistic regression analysis were included in the protein combination. For the derivation of protein signatures, logistic regression coefficients for each protein were identified and weighted according to protein expression values. Candidate protein signatures (AUC > 0.85, accuracy > 90%, *p* < 0.05) were validated by k-fold cross-validation to identify the optimal protein combination. The patients were randomly separated two-fold (training and test sets), 300 times.

### 2.7. Single-Cell RNA Sequencing Analysis

The right part of Figure 1 depicts a flow diagram of the scRNA-seq analysis. The analysis was performed using the GSE136103 dataset and the Seurat package version 4.0.5 (https://satijalab.org/seurat/index.html accessed on 1 January 2022). Pre-processing followed the GSE136103 method. For shared nearest neighbor clustering, variable features were determined using the variance stabilizing transformation (VST) of the selection method and 2000 variable counts. After scaling the data, a principal component analysis (PCA) was performed using the identified variable features. With the analyzed principal components, the optimal dimensions were analyzed using an elbow plot. According to the determined dimensions and principal components, single cells were clustered and visualized as a t-distributed stochastic neighbor embedding (t-SNE) graph. Clusters were identified using markers used to divide the clusters. Primary clustering was performed using data from five healthy and five cirrhotic patients. Secondary clustering was performed using cells presumed to be MPs. To identify the distribution in which cells express genes that code for differentially expressed proteins detected by DSP, the corresponding genes were marked in the secondary cluster using SCINA (semi-supervised category identification and assignment).

### 2.8. Statistical Analysis

Continuous data are presented in means with a standardized deviation and categorial variables are expressed as number and percentage. The categorial variables between fibrosis stage and clinicopathological variables were assessed using the chi-squared test or Fisher’s exact test and continuous variables between groups were evaluated using a Wilcoxon rank-sum test. The predictive accuracy of the threshold values for classifying fibrosis stages 0–2 and stages 3 and 4 was assessed using a receiver operating characteristic (ROC) curve analysis. The independence of the protein signature was analyzed using a logistic regression analysis of the protein signature and clinicopathological variables. Variables with *p* values less than 0.1 were included for the multivariate logistic regression analysis. The statistical significance was set at *p* < 0.05 (two-tailed). All statistical analyses were performed using R version 3.3.3 (R Development Core Team, https://www.r-project.org/ accessed on 16 February 2022).

## 3. Results

### 3.1. Patient Characteristics

Of the 88 ROIs, 64 with a low CD3 expression were presumed to be “non-inflammatory” regions, whereas 24 ROIs with a high CD3 expression were classified as “inflammatory” regions (Figure 1). The baseline characteristics of the enrolled patients are shown in Table 1. A total of 31 ROIs from the enrolled patients were classified as having early-stage fibrosis (seven as F0, 17 as F1, and seven as F2), and 33 ROIs were classified as having advanced fibrosis (15 as F3 and 18 as F4). The enrolled patients were predominantly male, and the mean ages were 54.23 and 50.91 years for patients with fibrosis stages 0–2 or stages 3–4, respectively. The most common etiology of chronic liver disease was CHB, which accounted for 67.74% and 75.76% of patients with fibrosis stages 0–2 or stage 3–4, respectively. Because nucleotide/side analogs for HBV therapy are reimbursed for patients with HBV DNA > 2000 IU/mL during cirrhosis in Korea, a larger proportion of patients with advanced fibrosis received antiviral therapy compared to those with early fibrosis (9.68% vs. 24.24%, *p* = 0.2255). There were no significant differences between the early and advanced fibrosis groups regarding albumin, AST, ALT, and platelet levels.

### 3.2. Multiplexed DSP of Protein Expression Level according to the Fibrosis Stage

Representative ROIs stained with CD3, CD68, and SMA according to the fibrosis stage are depicted in Figure 2A. A volcano plot for protein marker expression is depicted in Figure 2B. The figure compares protein expression levels in early (F0, F1, and F2) and advanced (F3 and F4) fibrosis. The results are summarized in Table 2. Moreover, CD68 and HLA-DR were upregulated 1.50-fold and 1.32-fold higher in advanced fibrosis compared to early-stage fibrosis. In contrast, protein markers other than CD68 and HLA-DR, such as phospho-c-RAF, stimulator of interferon genes (STING), OX40 ligand (OX40L), V-domain IgG suppressor of T cell activation (VISTA), pan RAS, and T cell immunoglobulin and mucin domain-containing protein 3 (TIM3) were downregulated in fibrosis stages 3 and 4, with fold changes of up to 2.65 (Table 2).

### 3.3. Protein Signatures for the Advanced Fibrosis Derived from the DSP Analysis

Using the DSP results from the CD68^+^ area, we identified unique immune-related protein signatures that reflect the advanced fibrosis. Table 3 shows the candidate protein signatures derived from the DSP. Of the five candidate protein signatures, one that was composed of the genes CD68, HLA-DR, OX40L, phospho-c-RAF, STING, and TIM3 showed the highest positive predictive value for advanced fibrosis, with an AUC in the ROC curve of 0.873 (0.791–0.955). The positive predictive value of the developed protein signatures was 93.10, and the negative predictive value was 82.86. The sensitivity and specificity of this protein signature were 81.82% and 93.55%, respectively (Table 3). Next, we validated the protein signature in the CHB group and non-CHB group. In the CHB subgroup, the AUC in the ROC curve of the protein signature was 0.846 (0.741–0.950) and the *p* value for the logistic regression analysis was 2.53 × 10^−4^. In the non-CHB subgroup, AUC in the ROC curve was 0.950 (0.849–1.000) and the *p* value for the logistic regression analysis was 0.052 (Appendix A)

We also evaluated the factors associated with high-grade fibrosis using a logistic regression analysis (Table 4). In the univariate analysis, the protein signature was the only factor associated with predicting advanced fibrosis (*p* = 1.16 × 10^–6^). The protein signature, age, and BMI were then included in the multivariate analysis. In the multivariate analysis, the protein signature (odds ratio = 104.13, 95% CI: 14.29–758.66, *p* = 4.54 × 10^–6^) and BMI (≤25 kg/m^2^ vs. >25 kg/m^2^) (odds ratio = 8.15, CI: 1.28–51.93, *p* = 0.026) were found to be significantly associated with advanced fibrosis. A multivariate analysis using the other four protein signatures is demonstrated in Appendix A.

### 3.4. Predicting Related Immune Cells in Different Fibrosis Stage Using Single Cell RNA Sequencing Database

Next, we tried to visualize our marker proteins from the DSP results in a t-SNE map derived from a publicly available scRNA-seq dataset (GSE136103). By utilizing the non-parenchymal liver cell atlas established by Ramachandran et al. [12], we have marked our DSP driven proteins into the atlas to specify the upregulated subpopulation of MPs. We re-analyzed the dataset and classified MPs into ten clusters, as depicted in Figure 3A. Each cluster showed high similarities to the t-SNE map previously presented by Ramachandran et al. [12]. Most of the protein markers that were upregulated in MPs of F0–2 by DSP were marked in clusters of tissue monocytes, as shown in the left panel of Figure 3B. Most of the protein markers that were upregulated in MPs of F3–4 by DSP were marked in clusters of SAMacs, KCs, and tissue monocytes (Figure 3B, right panel). The scaled gene expression in each cluster of MPs is shown in Figure 3C. CD68 and CD74, which were two genes that showed higher protein expression levels in advanced fibrosis in our DSP analysis, were also highly expressed in SAMacs1 and SAMacs2 rather than in tissue monocytes.

### 3.5. Validation of DSP Protein Analysis Using mRNA Expression Data from Snap-Frozen Livers Using NanoString nCounter MAX System

We also applied a NanoString nCounter MAX mRNA expression analysis using the same liver tissues (snap-frozen) that was used in the DSP analysis to validate our protein data. Figure 3D delineates the differentially expressed genes between fibrosis stage 0–2 and 3–4. A previous report demonstrated that TREM2 and CD9 are selectively upregulated in SAMacs and can be used as protein markers for this MP subset [12]. In our NanoString nCounter MAX mRNA expression analysis, TREM2, and CD9 were also shown to be more upregulated in F3–4 than in F0–2 (TREM2, fold change = 1.70, *p* = 0.16; CD9, fold change = 1.36, *p* = 5.54 × 10^−4^). Moreover, CD68, a protein that was newly identified to be upregulated in MPs in advanced fibrosis by a DSP analysis, was also significantly upregulated by the mRNA expression analysis (fold change = 1.29, *p* = 1.84 × 10^−4^). CD74, also known as HLA-DR antigens-associated invariant chain, showed a tendency of higher expression in advanced fibrosis compared to early fibrosis, although statistical significance was not met.

## 4. Discussion

In this study, using multiplexed DSP protein profiling and scRNA-seq database, we demonstrated the phenotypical heterogeneity of portal MPs according to the fibrosis stages for the first time. We have also identified a novel protein signature that predicted advanced fibrosis with a high reliability.

Liver fibrosis is a common pathological consequence of most chronic liver diseases. Fibrosis is associated with many NPCs, including inflammatory, endothelial, and mesenchymal cells. Numerous reports have elucidated the different phenotypes of NPCs depending on the presence or absence of liver fibrosis [21,22]. In addition, different roles of the MPs according to the fibrosis stage have been suggested in several previous studies [12]. KCs, which dominate the hepatic macrophage pool, are central to intrahepatic immunological tolerance and provide an anti-inflammatory micromilieu to the liver during homeostasis [9,23]. However, in acute or chronic liver injury, KCs secrete CCL2 and thereby recruit circulating monocytes to the liver, which then differentiate into MoMFs. These MoMFs prevail during liver injury and stimulate stellate cells [24]. This results in the excessive deposition of extracellular matrix, which leads to hepatic fibrosis. MacParland et al. mapped the cellular landscape of the human liver via scRNA-seq and reported that CD68^+^ macrophages have two distinct phenotypes that are classified as having pro-inflammatory or immune-regulatory roles [14]. Moreover, recent studies using scRNA-seq demonstrated that TREM^+^ CD9^+^ SAMacs were derived from circulating monocytes and demonstrated a pro-fibrogenic phenotype [12,25]. Collectively, these studies proposed distinct phenotypes of intrahepatic cell populations through scRNA-seq and suggested that the pro-inflammatory phenotype of an intrahepatic macrophage switches to the anti-inflammatory or pro-fibrogenic phenotype during the process of liver fibrosis.

To the best of our knowledge, this is the first study to show the different phenotypes of MPs between the early and late stages of liver fibrosis using spatially defined protein-based multiplexed profiling. The DSP transcriptome in our study was matched and analyzed using the publicly available RNA-seq dataset (GSE136103) from Ramachandran et al. [12]. Tissue monocytes appeared to be highly abundant in the portal area of livers with fibrosis stage 0–2, whereas KCs, SAMacs, and tissue monocytes were highly abundant in the portal area of livers with fibrosis stage 3 and 4. In addition, an mRNA expression analysis using an nCounter gene expression assay showed a higher expression level of representative markers of SAMacs, namely TREM2, CD68, and CD9 in advanced fibrosis, supporting the DSP results. These results are consistent with previous reports demonstrating the pro-fibrogenic phenotype of SAMacs, which are thought to be derived from circulating monocytes.

Herein, we also proposed a novel protein signature derived from DSP data for advanced liver fibrosis, which showed good performance regardless of the etiology of the liver disease. The proposed protein signature was composed of six different proteins: CD68, HLA-DR, OX40L, phospho-cRAF, STING, and TIM3. CD68 and HLA-DR were upregulated in fibrosis stages 3 and 4, whereas the other four proteins were downregulated in the advanced stages compared to stages 1 and 2. CD68, a type 1 transmembrane glycoprotein of 110 kDa, is a representative macrophage marker. A recent study using mice livers demonstrated an increase in CD68^+^ macrophages in advanced liver fibrosis or cirrhosis compared to a normal liver. It also found that CD68^+^ macrophages were predominantly concentrated in scars during advanced fibrosis, suggesting its pro-fibrogenic role in the process of liver fibrosis [4,26]. HLA-DR is routinely used to identify macrophage lineages, such as Kupffer cells and circulating macrophages, and is widely present in antigen-presenting cells [27]. It is known to be upregulated upon immune stimulation and is proposed to be a monocyte activation marker [28]. Our very recent study demonstrated a higher level of HLA-DR in intrahepatic monocytes in human livers with advanced fibrosis than that with early fibrosis [24]. OX40L, phospho-cRAF, STING, and Tim3, which are shown to be downregulated in the advanced stage of fibrosis, are known to be related to the inflammatory change of intrahepatic monocytes. OX40L, a member of the tumor necrosis factor superfamily, interacts with OX40 and is associated with the secretion of pro-inflammatory cytokines in the setting of non-alcoholic steatohepatitis in mice [29]. Raf kinase is thought to promote cell growth through the direct phosphorylation of mitogen-activated protein kinase (MAPK) and activation of its downstream signaling [30,31]. Recently, interleukin-9 (IL-9) is increased in liver cirrhosis and CHB with fibrosis [32]. IL-9 was also shown to be related to the activation of the Raf/MEK/ERK signaling pathway [33]. In terms of STING, it is an important innate immune protein that coordinates with multiple immune responses, including the induction of interferons [34,35]. In liver, STING is mainly expressed in NPCs, such as Kupffer cell, hepatic stellate cell, and sinusoidal endothelial cells [36]. Recently, STING activation is found to be associated with hepatic inflammation for various types of liver disease, including CHB, CHC, and non-alcoholic fatty liver disease [36]. Moreover, the role of STING on liver fibrosis has suggested its possible role as a therapeutic target for liver fibrosis [37,38,39]. Lastly, TIM3 is a surface marker for terminally differentiated T-cells, is also expressed in monocytes, and is thought to have a regulatory role in liver fibrosis [40]. The expression level of TIM3 in monocytes being decreased in cirrhosis also suggested that high levels of TIM3 blocks monocyte activation [41,42].

The protein signatures are tissue-driven biomarkers and, for that reason, are difficult to apply in everyday clinical practice. However, they can be considered as bridges in establishing liquid biopsy biomarkers to predict advanced fibrosis. Blood-based biomarkers may be identified by analyzing the miRNAs or gene signatures associated with our tissue-driven protein signatures. In addition, since the process of fibrosis is like a concerto of various immune cells, the role of various types of MPs on fibrogenesis is expected to be revealed more in depth. Recently, other various subsets of liver resident macrophages, such as liver capsular macrophages and lipid-associated macrophages, have been shown to play important roles in the process of liver inflammation [43,44]. Moreover, Wang et al. demonstrated a novel finding that peritoneal cavity macrophages migrate to the liver parenchyma in the setting of liver injury and suggested that they play an essential role in tissue repair [45]. However, these studies focused on mice models and not human subjects. Moreover, our work restricted ROIs only to the portal area to increase the probability of detecting immune markers. Consequently, the coordination of these macrophages in relation to fibrosis did not fall into our interest. Future studies using proposed markers for aforementioned macrophages are mandatory to unveil the harmony of intrahepatic macrophages in the process of liver fibrosis.

## 5. Conclusions

This is the first study that used spatially defined protein-based multiplexed profiling to show the critical difference in the phenotypes of portal MPs between livers with early- or late-stage fibrosis. The results were validated using internal gene expression data and a publicly available scRNA-seq database. Our findings developed a novel protein signature predicting advanced fibrosis with high reliability. Using DSP, we have specified the region of protein analysis to the periportal area where MPs are abundant. Further studies are essential to validate the proposed protein signature and to identify the mechanisms causing the phenotypical differences in these portal MP populations between different fibrosis stages.

## Figures and Tables

**Figure 1 cells-11-03387-f001:**
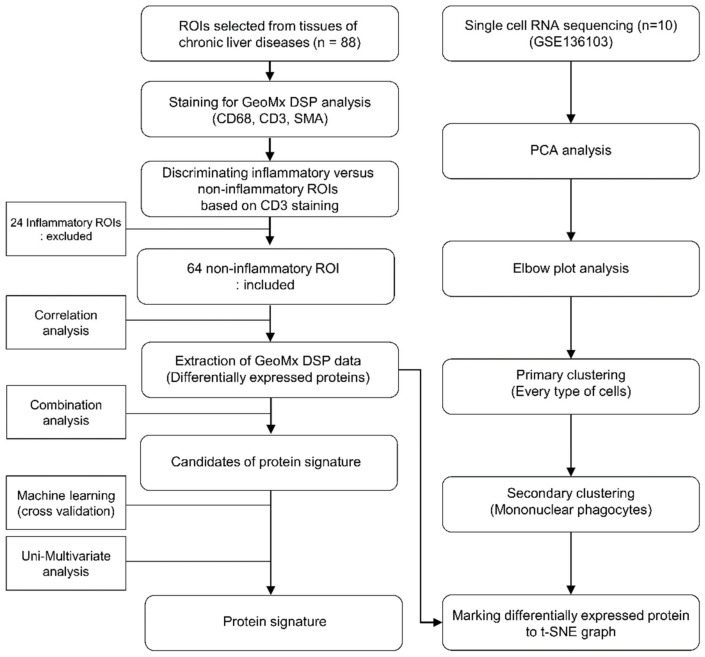
Flow chart of the DSP analysis and the single cell-DSP matching.

**Figure 2 cells-11-03387-f002:**
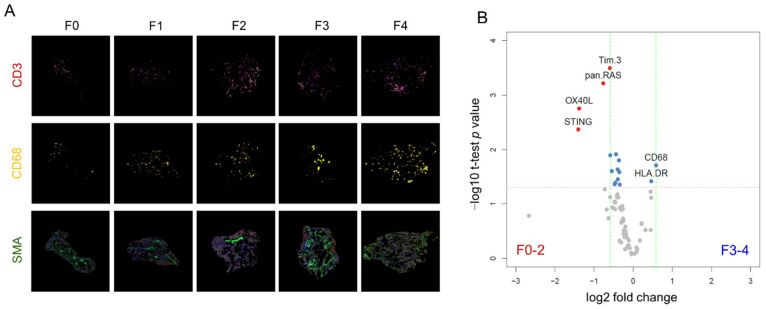
Multiplexed DSP of protein expression level according to the fibrosis stage. (**A**) Representative ROIs according to the fibrosis stage. Each core was stained with CD3 (red), CD68 (yellow), and SMA (green) antibodies. (**B**) Volcano plot describing the differentially expressed proteins in portal CD68^+^ areas between fibrosis stages 0–2 and stages 3–4. Proteins that are highly expressed in early-stage fibrosis are indicated by red dots (TIM3, pan-RAS, OX40L, and STING), whereas proteins that are highly expressed in advanced stage fibrosis are indicated by blue dots (HLA-DR and CD68).

**Figure 3 cells-11-03387-f003:**
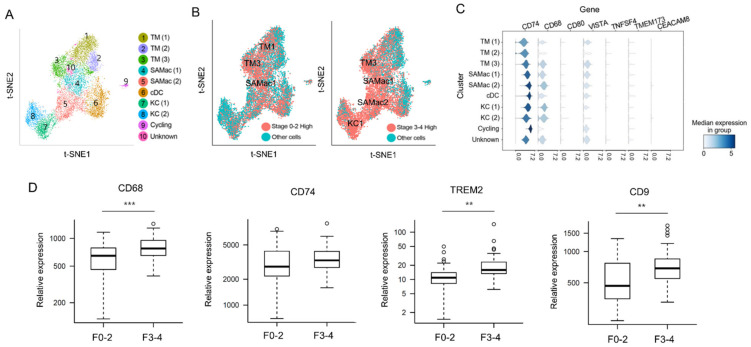
Combined analysis of scRNA-seq with DSP. (**A**) Reconstituted t-SNE graph for intrahepatic MPs using GEO datasets (GSE136103). (**B**) Marking differentially expressed proteins between early-stage fibrosis and advanced-stage fibrosis from DSP data to reconstituted t-SNE graph. Left panel is the graph that marked proteins highly expressed in F0–2, and right panel is the graph that marked proteins highly expressed in F3–4. (**C**) Scaled gene expression across every cluster of MPs. (**D**) Gene expression analysis with snap-frozen liver tissues using NanoString nCounter expression analysis. The expression levels of CD68, CD74, TREM2, and CD9 were compared between livers with early fibrosis and advanced fibrosis. Significance was indicated with ** *p* value < 0.01, *** *p* value < 0.001.

**Table 1 cells-11-03387-t001:** Baseline characteristic of enrolled patients.

	Fibrosis Stage 0–2 (N = 31)	Fibrosis Stage 3–4 (N = 33)	*p* Value **
Sex			0.322 #
Male	20 (64.5)	26 (78.8)	
Female	11 (35.5)	7 (21.2)	
Mean age (±SD)	54.2 (±11.3)	50.9 (±8.2)	0.186 $
Etiology			0.586 ^
CHB	21 (67.7)	25 (75.8)	
CHC	2 (6.5)	0 (0.0)	
Alcohol	3 (9.7)	2 (6.1)	
Others	5 (16.1)	6 (18.2)	
Diabetes	6 (19.4)	11 (33.3)	0.326 #
BMI			0.085 #
≤25 kg/m^2^	25 (80.7)	19 (57.6)	
>25 kg/m^2^	6 (19.4)	14 (42.4)	
ALT			1.000 #
≥31 (F), ≥41 (M) IU/L	12 (38.7)	13 (39.4)	
AST			1.000 #
≥31 (F), ≥37 (M) IU/L	19 (61.3)	20 (60.6)	
GGT	(−5)	(−1)	0.640 #
≥50 (IU/L)	18 (58.1)	25 (78.1)	
Albumin			0.644 #
<4.0 g/dL	14 (45.2)	12 (36.4)	
Platelets			0.543 #
<150 × 10^9^/L	9 (29.0)	13 (39.4)	
Cholesterol (mmol/L)	(−2)		0.942 #
≥200 mg/dL	5 (16.1)	7 (21.2)	
Antiviral treatment	3 (9.7)	8 (24.2)	0.226 #
Fibrosis			
Stage 0	7 (22.6)	0 (0.0)	
Stage 1	17 (54.8)	0 (0.0)	
Stage 2	7 (22.6)	0 (0.0)	
Stage 3	0 (0.0)	15 (45.5)	
Stage 4	0 (0.0)	18 (54.6)	

Data are presented as N (%), mean ± SD. ALT, alanine aminotransferase; AST, aspartate aminotransferase; BMI, body mass index; chronic hepatitis B, CHB; chronic hepatitis C, CHC; gamma glutamyl transferase, GGT; SD, standard deviation. # Chi squared test; ^ Fisher’s exact test; $ Student’s *t*-test. ** *p*-value < 0.01.

**Table 2 cells-11-03387-t002:** List of the differentially expressed proteins according to the fibrosis stages (F0–2 vs. F3–4).

SEQ	Protein	N	Coef	Logistic Regression *p*-Value	Wilcoxon Test *p*-Value	Fold Change	F0–2 (N = 31)	F3–4 (N = 33)
1	CD68	64	0.00650	2.78 × 10^−2^	1.96 × 10^−2^	1.50	118.62	178.04
2	HLA.DR	64	0.02140	4.62 × 10^−2^	3.83 × 10^−2^	1.38	35.41	48.82
3	Phospho.c.RAF	64	−1.06091	4.96 × 10^−2^	4.43 × 10^−2^	−1.27	1.23	0.97
4	Cleaved.Caspase.9	64	−0.10881	3.22 × 10^−2^	2.59 × 10^−2^	−1.28	14.62	11.46
5	CD127	64	−0.48734	2.09 × 10^−2^	1.57 × 10^−2^	−1.29	3.77	2.93
6	ARG1	64	−0.00693	4.21 × 10^−2^	3.52 × 10^−2^	−1.31	200.13	152.53
7	Beta.2.microglobulin	64	−0.11174	2.84 × 10^−2^	2.32 × 10^−2^	−1.31	12.67	9.63
8	X4.1BB	64	−1.28366	1.77 × 10^−2^	1.22 × 10^−2^	−1.35	1.41	1.04
9	LAG3	64	−0.94575	4.71 × 10^−2^	4.05 × 10^−2^	−1.37	1.17	0.85
10	B7.H3	64	−0.07086	3.41 × 10^−2^	2.50 × 10^−2^	−1.46	19.21	13.17
11	VISTA	64	−0.41498	2.02 × 10^−2^	1.27 × 10^−2^	−1.50	3.58	2.39
12	Tim.3	64	−0.50608	1.91 × 10^−3^	3.18 × 10^−4^	−1.51	5.56	3.68
13	pan.RAS	64	−1.75375	1.46 × 10^−3^	6.02 × 10^−4^	−1.70	1.69	1.00
14	OX40L	64	−1.13933	8.75 × 10^−3^	1.76 × 10^−3^	−2.60	2.69	1.03
15	STING	64	−0.09585	8.76 × 10^−3^	4.27 × 10^−3^	−2.65	19.19	7.26

Abbreviation: CD68, cluster of differentiation 68; HLA.DR, human leukocyte antigen-DR isotype; ARG1, arginase 1; LAG3, lymphocyte activation gene 3; B7.H3, B7 homolog 3 protein; VISTA, V-domain IgG suppressor of T cell activation; Tim.3, T cell immunoglobulin and mucin domain-containing protein 3; OX40L, OX40 ligand; STING, stimulator of interferon genes.

**Table 3 cells-11-03387-t003:** Candidate protein signatures derived from DSP.

SEQ	Protein	*p*_Value	AUROC	Sensitivity	Specificity	Accuracy	PPV	NPV
1	CD68_HLA.DR_OX40L_Phospho.c.RAF_STING_Tim.3	1.16 × 10^−6^	0.873	81.82	93.55	87.50	93.10	82.86
2	ARG1_B7.H3_CD127_CD68_HLA.DR_OX40L_pan.RAS_STING_Tim.3	5.19 × 10^−7^	0.894	87.88	83.87	85.94	85.29	86.67
3	B7.H3_CD68_HLA.DR_OX40L_Phospho.c.RAF_STING_VISTA	5.19 × 10^−7^	0.874	87.88	83.87	85.94	85.29	86.67
4	Beta.2.microglobulin_CD127_CD68_HLA.DR_OX40L_Tim.3	5.06 × 10^−7^	0.878	84.85	87.10	85.94	87.50	84.38
5	CD68_HLA.DR_OX40L_pan.RAS_STING	5.06 × 10^−7^	0.870	84.85	87.10	85.94	87.50	84.38

Abbreviation: AUROC, area under receiver operating characteristic curve; NPV, negative predictive value; PPV, positive predictive value; SEQ, sequence.

**Table 4 cells-11-03387-t004:** Uni/multi-variate logistic regression analysis of factors associated with high-grade fibrosis.

Univariable Logistic Regression					
Variable	*n*	Coefficient	Se (Coefficient)	Odds Ratio (95% CI)	*p*-Value
CD68_HLA-DR_OX40L_Phospho-c-RAF_STING_Tim-3 (low vs. high)	64	4.178	0.859	65.25 (12.11–351.49)	1.16 × 10^−6^
Age (≤55 years vs. >55 years)	64	−0.945	0.543	0.39 (0.13–1.13)	0.082
Sex (male vs. female)	64	−0.714	0.568	0.49 (0.16–1.49)	0.208
**Etiology**					
CHB (absent vs. present)	64	0.397	0.559	1.49 (0.50–4.45)	0.477
CHC (absent vs. present)	64	−16.695	1696.734	0.00 (0.00–Inf)	0.992
Alcohol (absent vs. present)	64	−0.507	0.949	0.60 (0.09–3.87)	0.593
Others (absent vs. present)	64	0.145	0.665	1.16 (0.31–4.25)	0.828
BMI (≤25 kg/m^2^ vs. >25 kg/m^2^)	64	1.122	0.575	3.07 (0.99–9.48)	0.051
Diabetes (absent vs. present)	64	0.734	0.586	2.08 (0.66–6.57)	0.210
ALT (<31(F), <41(M) IU/L vs. ≥31(F), ≥41(M) IU/L)	64	0.029	0.513	1.03 (0.38–2.81)	0.955
AST (<31(F), <37(M) IU/L vs. ≥31(F), ≥37(M) IU/L)	64	−0.029	0.513	0.97 (0.36–2.65)	0.955
**GGT (<50 IU/L vs.** **≥** **50 IU/L)**	58	0.462	0.603	1.59 (0.49–5.17)	0.443
**Albumin (<4.0 g/dL vs.** **≥** **4.0 g/dL)**	64	0.366	0.511	1.44 (0.53–3.92)	0.475
Platelets (<150 × 10^9^/L vs. ≥150 × 10^9^/L)	64	−0.463	0.532	0.63 (0.22–1.79)	0.384
Cholesterol (<200 mg/dL vs. ≥200 mg/dL)	62	0.256	0.650	1.29 (0.36–4.62)	0.693
**Multivariable Logistic Regression**					
**Variable**	**Coefficient**		**Odds Ratio (95% CI)**		** *p* ** **-Value**
CD68_HLA-DR_OX40L_Phospho-c-RAF_STING_Tim-3 (Low vs. High)	4.646		104.13 (14.29–758.66)		4.54 × 10^−6^
Age (≤55 years vs. >55 years)	−0.558		0.57 (0.10–3.29)		0.532
BMI (≤25 kg/m^2^ vs. >25 kg/m^2^)	2.097		8.15 (1.28–51.93)		0.026

Abbreviation: ALT, alanine aminotransferase; AST, aspartate aminotransferase; body mass index, BMI; chronic hepatitis B, CHB; chronic hepatitis C, CHC; CI, confidence interval; gamma glutamyl transferase, GGT.

## Data Availability

The datasets generated during and/or analyzed during the current study are available from the corresponding author on reasonable request.

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
