# Peer review of "Multiplexed Digital Spatial Protein Profiling Reveals Distinct Phenotypes of Mononuclear Phagocytes in Livers with Advanced Fibrosis"

_cells, 2022, doi:10.3390/cells11213387_

Round 1

Reviewer 1 Report

1. The authors presented very interesting data, in connection with which several questions arose. Not so long ago, special populations of macrophages were found in the liver, in addition to the well-known Kupffer cells. First of all, these are macrophages associated with the liver capsule (https://pubmed.ncbi.nlm.nih.gov/28813662/), as well as around ductal macrophages, which have specific phenotypes (https://pubmed.ncbi.nlm.nih.gov/32888418/). Is it possible to detect the participation of these populations of macrophages in the development of liver fibrosis using the research methods used by the authors? And if the authors do not see their participation in the development of liver fibrosis, then what could be the reason for this?

2. Since the authors describe populations of liver macrophages, in my opinion, it is worth mentioning the macrophages of the capsule, macrophages associated with the bile ducts, as well as the possible involvement of peritoneal macrophages in maintaining the size of the general population of liver macrophages (https://pubmed.ncbi.nlm.nih.gov/27062926/).

3.  It is necessary to decipher the abbreviation  SAMacs  in the abstract of the article.

Author Response

  1. The authors presented very interesting data, in connection with which several questions arose. Not so long ago, special populations of macrophages were found in the liver, in addition to the well-known Kupffer cells. First of all, these are macrophages associated with the liver capsule (https://pubmed.ncbi.nlm.nih.gov/28813662/, sierro et al.), as well as around ductal macrophages, which have specific phenotypes (https://pubmed.ncbi.nlm.nih.gov/32888418/, remmerie et al. ). Is it possible to detect the participation of these populations of macrophages in the development of liver fibrosis using the research methods used by the authors? And if the authors do not see their participation in the development of liver fibrosis, then what could be the reason for this?

Response: We are grateful for the positive comments from the Reviewer. We really appreciate this professional and valuable recommendation. We have made every effort to address the issues raised and to respond to all comments carefully. The revisions are indicated in red font in the revised manuscript. A detailed point-by-point response to the reviewer's comments are given below. We hope that our revisions would meet the reviewer’s expectations.

As the reviewer has suggested, many different types of macrophages are present including SAMacs, Kupffer cells, etc. However, in this study, we only focused on the portal area and have specified the region of interest only to the portal areas to increase the probability of detecting immune cell markers. Therefore, it did not fall into our interest to evaluate the participation of liver capsular macrophages.

Moreover, concerning the fact that the above-mentioned studies used only mouse models and not human subjects, we believe that their study designs differ significantly from our own. We have added this in Discussion in line 414-427 as follows:

“In addition, since the process of fibrosis is like a concerto of various immune cells, the role of various types of MPs on fibrogenesis is expected to be revealed more in depth. Recently, other various subsets of liver resident macrophages, such as liver capsular macrophages and lipid-associated macrophages have been shown to play important roles in the process of liver inflammation [43,44]. Moreover, Wang et al. demonstrated a novel finding that peritoneal cavity macrophages migrate to the liver parenchyma in the setting of liver injury and suggested that they play an essential role in tissue repair [45]. However, these studies focused on mice models and not human subjects. Moreover, our work restricted ROIs only to the portal area to increase the probability of detecting immune markers. Consequently, coordination of these macrophages in relation to fibrosis did not fall into our interest. Future studies using proposed markers for aforementioned macrophages is mandatory to unveil the harmony of intrahepatic macrophages in the process of liver fibrosis.”

We also added these interesting studies in our reference section as references #43 and 44.

  1. Since the authors describe populations of liver macrophages, in my opinion, it is worth mentioning the macrophages of the capsule, macrophages associated with the bile ducts, as well as the possible involvement of peritoneal macrophages in maintaining the size of the general population of liver macrophages (https://pubmed.ncbi.nlm.nih.gov/27062926/, Wang et al. ).

Response: Thank you for introducing us such an interesting article. We have added following sentences in Discussion and also added this study in References as reference number #45. 

……Moreover, Wang et al. demonstrated a novel finding that peritoneal cavity macrophages migrate to the liver parenchyma in the setting of liver injury and suggested that they play an essential role in tissue repair [45]. However, these studies focused on mice models and not human subjects. Moreover, our work restricted ROIs only to the portal area to increase the probability of detecting immune markers. Consequently, coordination of these macrophages in relation to fibrosis did not fall into our interest. Future studies using proposed markers for aforementioned macrophages is mandatory to unveil the harmony of intrahepatic macrophages in the process of liver fibrosis.…..

  1. It is necessary to decipher the abbreviation SAMacs in the abstract of the article.

Response: Thank you for the valuable comment. We have revised the manuscript accordingly.

Author Response

Jaejun Lee and colleagues present results from a retrospective study on portal proteins in individuals with liver biopsy proven chronic liver disease. They demonstrate a different phenotype between early- and latestage liver fibrosis and suggest a novel protein signature to predict advanced liver fibrosis. The study is well conducted, and results are well presented and discussed.

Response: Thank you for the helpful comments to improve the quality of manuscript. We have made every effort to address the issues raised and to respond to all comments. The revisions are indicated in red font in the revised manuscript. Please find next, a detailed, point-by-point response to the reviewer's comments. We hope that our revisions would meet the reviewer’s expectations.

Materials and Methods

P3, L111-120: Please add a few more details in the Patients section to better understand the study population. E.g., where is Ajou Medical Center located? How was “chronic liver disease” defined? Was it defined by ICD-10 coding, and if so, please add which codes were used?

Response: Thank you for the helpful comment. According, we have added whereabout of Ajou Medical Center and the definition of chronic liver disease in line 114,116-120 as follows;

“Chronic liver disease was defined according to the recently introduced EASL guideline as follows: decreased liver function caused by chronic inflammation from any source, including chronic hepatitis B and C, alcoholic and non-alcoholic fatty liver disease, and other etiologies that can cause chronic liver inflammation.”

P5, L185 – 186: How was variables for the uni – and multivariate logistic regression analysis selected? A priori based on current knowledge or based on p-values? Why was only one of the protein signatures tested in the logistic regression analysis? Please clarify this and consider testing the other protein signatures as well. Further, it would be very interesting to also include result on the other etiologies “alcohol” and “others”, as these patients represent a large proportion of the study population and may confound the effect of the protein signature.

Response: Thank you for the critical comments. We have added methodological details in line 197–198 in the Statistical analysis section accordingly. Etiologies including alcohol and others are now included for univariate logistic regression analysis and appeared to be insignificant in terms of correlation with liver fibrosis. (P value = 0.5930 and 0.8280, alcohol and other, respectively). Additional analysis has been added to Table 4. We have also evaluated other protein signatures for multivariate regression analysis and the results are presented in Supplementary table 1.

Results

P5, L193: How is “low CD3 expression defined”? Please clarify in the methods section

Response: Thank you for the comment. In lines 154-158, we added a section ‘Selecting ROIs for establishing the protein signature’ and answered question raised by the reviewer as follows:

“Selecting ROIs for establishing the protein signatures

CD3 expression levels were determined using the fluorescence intensity observed in GeoMx analysis and the median value was used to classify the ROIs into “inflammatory” and “non-inflammatory”. We excluded inflammatory ROIs because the phenotypes of the immune cells may not actually reflect the fibrogenesis process but the liver injury process.”

P5, L195 – 198: The section “CD3 expression levels were determined … the liver injury process.” should be moved to the Methods section.

Response: We have moved the sentence to Material and Methods in line 154–158 accordingly.

P5, L204: How was hepatitis B infection defined? Same for hepatitis C infection.

Response: We have added relevant information to Materials and Methods, line 120-122 as follows:

“Chronic hepatitis B (CHB) was defined as presence of hepatitis B surface antigen for more than 6 months, and chronic hepatitis C (CHC) infection was defined as the presence of HCV RNA for more than 6 months.”

P8, L248 – 259: Suggest deleting the phrase “There were liver tissues …” as these numbers are presented in table 1.

Response: We have deleted the sentence as per the reviewer’s suggestion.

P8, L252 - 253: The sentence “Compared to early …” is difficult to understand. Please rephrase.

Response: In line 250-252, we have rephrased the sentences as the followings:

“CD68 and HLA-DR were upregulated 1.50-fold and 1.32-fold higher in advanced fibrosis compared to early-stage fibrosis”

P9, L275 – 282: Please avoid repeating results presented in Table 3.

Response: We have reduced the paragraph by deleting the duplicate contents.

P9, L285 - 292: Please delete the sentence “Variables such as …” as this should be put in the statistics section. What is an “acceptable p-value”. Please clarify and move to the statistics section

Response: We have removed the sentence accordingly. In the revised manuscript, line 197-198, we have added sentence explaining the p value for multivariate analysis as follows:

“Variables with p values less than 0.1 were included for multivariate logistic regression analysis.”

Table 1: Suggest shortening this table by deleting either “yes” or “no” rows. Likewise, both intervals for results on liver enzymes are not needed. Number of individuals with HBV and HCV are reported twice. Suggest deleting the results from the first part of the table and report it under the sub-headline “Etiology” Results reported on GGT levels does not match N=31 and N=33. Please update the numbers.

Response: We have revised Table 1 accordingly. Regarding GGT levels, we had only 58 ROIs with information on GGT levels. Therefore, the total number was slightly different from other variables. In addition, we have corrected numbers on GGT levels as they were entered incorrectly by mistake.

Table 3: It is very difficult to read this table, please reformat and please add footnote on the abbreviations used.

Response: We have revised in accordance with the reviewer’s suggestion.

Table 4: Consider to also report odds ratios for the univariate analysis, which will help the interpretation of the results.

Response: We agree with the reviewer, that presenting odds ratio will help readers to better understand the table. We have revised the Table 4 accordingly.

Discussion

P13, L392 – 414: In this part of the discussion, results on protein signatures are compared with previous literature and especially literature on NASH/NAFLD. However, it is not clear whether NASH/NAFLD patients are included in this study population and thus, are these perspectives relevant for this study? One may think that the category of “others” includes NASH/NAFLD patient, but this is not clear from the presented data. In contrast, literature on HBV/HCV and alcohol are not discussed. What does not literature show in these patients? Do the protein signatures vary in different patient groups? Consider including “alcohol” and “others” in the analysis, results, and discussion to be able to distinguish between patient groups even though numbers are small. Alternatively, focus less on the etiologies in the discussion section.

Response: In the revised manuscript, we have added references which could explicate the role of STING and Raf kinase in various types of liver diseases. Also, we have deleted references that could overemphasize the etiology of NASH in this section. In line 397-406, we have added and revised the sentences and references (#32, #33, #36, #38, #39) as follows:

“Raf kinase is thought to promote cell growth through the direct phosphorylation of mitogen-activated protein kinase (MAPK) and activation of its downstream signaling [30,31]. Recently, interleukin-9 (IL-9) is increased in liver cirrhosis and CHB with fibrosis [32]. IL-9 was also shown to be related to activation of the Raf/MEK/ERK signaling pathway [33]. In terms of STING, it is an important innate immune protein that coordinates with multiple immune responses, including the induction of interferons [34,35]. In liver, STING is mainly expressed in NPCs such as Kupffer cell, hepatic stellate cell, and sinusoidal endothelial cells [36]. Recently, STING activation is found associated with hepatic inflammation for various types of liver disease, including CHB, CHC, and non-alcoholic fatty liver disease [36]. Moreover, the role of STING on liver fibrosis and has suggested its possible role as a therapeutic target for liver fibrosis [37–39].”

References

  1. Qin, S.-y., et al., A deleterious role for Th9/IL-9 in hepatic fibrogenesis. Scientific Reports, 2016. 6(1): p. 18694.
  2. Guo, X., et al., CXCL10-induced IL-9 promotes liver fibrosis via Raf/MEK/ERK signaling pathway. Biomedicine & Pharmacotherapy, 2018. 105: p. 282-289.
  3. Chen, C., R.X. Yang, and H.G. Xu, STING and liver disease. J Gastroenterol, 2021. 56(8): p. 704-712.
  4. Iracheta-Vellve, A., et al., Endoplasmic Reticulum Stress-induced Hepatocellular Death Pathways Mediate Liver Injury and Fibrosis via Stimulator of Interferon Genes. J Biol Chem, 2016. 291(52): p. 26794-26805.
  5. Li, Y., et al., STING signaling activation inhibits HBV replication and attenuates the severity of liver injury and HBV-induced fibrosis. Cell Mol Immunol, 2022. 19(1): p. 92-107.

Moreover, we made a subgroup analysis for HBV population and validated the protein markers in HBV and non-HBV subgroup (supplementary figure 1). It showed that the protein signature was highly sensitive and specific in detecting advanced fibrosis in both HBV group and non-HBV group (AUROC=0.846 (0.741–0.950), AUROC=0.950 (0.849-1.000), HBV and non-HBV, respectively). It shows that performance of the protein signature does not vary by the presence of HBV, demonstrating that the protein signature is applicable regardless of the etiology of liver disease.

We have added sentences concerning this matter in line 278-282 as follows:

“Next, we validated the protein signature in the CHB group and non-CHB group. In the CHB subgroup, the AUC in the ROC curve of the protein signature was 0.846 (0.741–0.950) and the p value for logistic regression analysis was 2.53×10−4. In the non-CHB subgroup, AUC in the ROC curve was 0.950 (0.849–1.000) and the p value for logistic regression analysis was 0.052. (Supplement figure 1).

Minor comments:

P2, L87: Please add a reference after “… in the fibrotic liver while fibrosis proceeds.”.

Response: We thank the Reviewer for giving us an opportunity to clarify this issue. In the revised manuscript line 85–87, we slightly changed the sentence to avoid confusion and added two relevant references. The revised sentences and references added are as follows. 

“SAMacs are reported to play a critical role in the initiation, progression and/or resolution of liver fibrosis and continue to play a role in the fibrotic liver while fibrosis proceeds.”

References

  1. Fallowfield, J.A., et al., Scar-Associated Macrophages Are a Major Source of Hepatic Matrix Metalloproteinase-13 and Facilitate the Resolution of Murine Hepatic Fibrosis. The Journal of Immunology, 2007. 178(8): p. 5288-5295.
  2. Ramachandran, P., et al., Resolving the fibrotic niche of human liver cirrhosis at single-cell level. Nature, 2019. 575(7783): p. 512-518.

Please reduce the number of decimals to one digit in e.g., ages and consider reducing the number of decimals reported throughout the manuscript.

Response: Thank you for the helpful comment. We have revised Table 1 and reduced the number of decimals to one digit. Furthermore, we also reduced the number of decimals in Table 2 and Table 4 to improve the readability. 

Round 2

Reviewer 2 Report

Dear Dr. Lee and Colleagues,

Thank you for the revised manuscript, which has improved significantly. 

The results on HBV are very interesting and make the results even more robust. 

One last comment:

There seems to be a discrepancy between the referencelist of the revised manuscript and the references mentioned in the point-by-point respons. Please make sure that the reference list are correct. 

Author Response

We have cross-checked the manuscript to ensure that citation numbers exactly matched the reference lists in the manuscript. Reference list is now correct. Thank you for the helpful comment.